# SeisMark: A Large-Scale Open Benchmark for Robust 3D Seismic Fault Detection

**Min Jun Park** [1]   **Joseph Stitt** [1]   **Robert Graham Clapp** [1]   **Ilan Naiman** [1]   **Artem Goncharuk** [1]   **Kevin F. Smith** [1]

## Abstract

We introduce SeisMark, a large-scale open benchmark designed to bridge the gap between verifiable ground truth and realistic texture in 3D seismic fault detection. Using a novel pipeline merging procedural geology with diffusion-based synthesis, we produce domain-realistic (survey-specific) textured volumes that expose significant brittleness in existing models masked by simplified physics data. Experiments demonstrate that SeisMark acts as a rigorous discriminator, distinguishing robust modern architecture from legacy model that suffers performance collapse under realistic domain shifts. We release this benchmark to the community to serve as a verifiable standard for developing trustworthy, deployment-ready AI for safety-critical subsurface applications.

## 1. Introduction

Subsurface characterization constitutes a cornerstone of modern energy transition strategies. As the demand for Carbon Capture and Storage (CCS) and geothermal energy grows, the ability to map the Earth's interior with precision becomes safety-critical. Among geological features, faults—fractures where rock layers have been displaced—are particularly consequential. In CCS projects, faults can act as "thief zones" or leakage pathways for injected $CO_2$, compromising long-term seal integrity (Alcalde et al., 2018; Ringrose & Meckel, 2019). Consequently, accurate fault mapping transcends simple exploration; it is a prerequisite for securing gigaton-scale storage and ensuring geothermal stability.

To map these deep subsurface structures, the industry relies on 3D seismic reflection surveys. Often described as an ultrasound of the Earth, this method records the echo of acoustic waves bouncing off rock layers to image the subsurface.

[1]X, the Moonshot Factory, Mountain View, CA, USA. Correspondence to: Min Jun Park <minjunp@google.com>.

*Proceedings of the 43$^{rd}$ International Conference on Machine Learning*, Seoul, South Korea. PMLR 306, 2026. Copyright 2026 by the author(s).

From a machine learning perspective, the resulting data manifests as massive volumetric tensors ($D \times H \times W$), often spanning millions to billions of voxels. While analogous to 3D medical imaging, seismic volumes are plagued by significantly lower signal-to-noise ratios, acquisition artifacts, and complex, non-stationary textures arising from geologic complexity. In this context, fault detection becomes a challenging 3D semantic segmentation task: identifying sparse, planar discontinuities embedded within chaotic, noisy strata.

Progress in automating this task is stifled by a fundamental data paradox: the absence of a standardized, verifiable benchmark. Unlike computer vision, where progress was accelerated by crowdsourced benchmarks like ImageNet (Deng et al., 2009), geophysics lacks accessible ground truth; structures buried kilometers deep cannot be physically verified. Researchers are thus forced to rely on scarce public field data, such as the F3 Netherlands survey (Silva et al., 2019). However, reliance on such data is fraught with challenges. First, field data labels are subjective, relying on human expert interpretation which often varies significantly between interpreters (Bond et al., 2007); this is particularly problematic with small faults which often go unlabeled. Second, the scarcity of public volumes creates a high risk of overfitting to specific geological contexts, preventing models from generalizing to diverse global geologies.

To circumvent the scarcity of reliable labels, the community has increasingly turned to synthetic data generation (Fehler & Keliher, 2011; Vizeu et al., 2022; Noori et al., 2020; Wu et al., 2019). This approach offers significant theoretical advantages: it grants researchers the freedom to simulate arbitrary geological structures—ranging from simple layering to complex folding—and inherently provides pixel-perfect ground truth labels. However, current synthetic workflows often necessitate a trade-off between structural diversity and textural realism. Standard physics-based simulations typically rely on simplified acoustic approximations that fail to capture the chaotic noise, multiples, and acquisition footprints inherent in real-world surveys. This deficiency creates a significant "Sim2Real" gap, meaning that existing synthetic datasets serve as poor proxies for real-world complexity, limiting their utility for rigorous evaluation in safety-critical subsurface applications (Zhou et al., 2021;

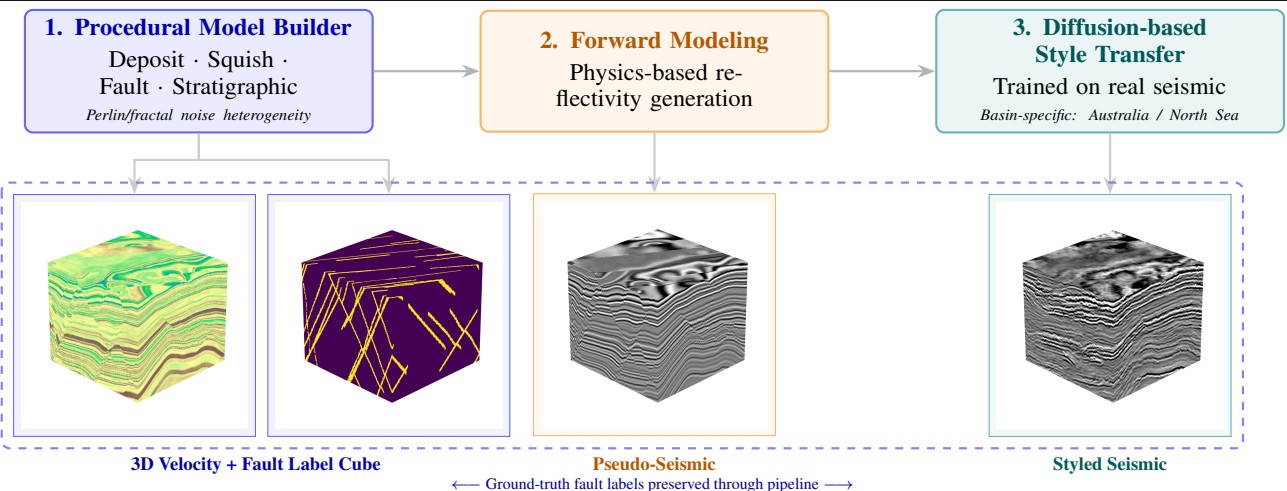

*Figure 1.* Synthetic seismic data generation pipeline. Stage 1 builds procedural 3D velocity models with accompanying ground-truth fault label cubes. Stage 2 converts velocity to pseudo-seismic via physics-based reflectivity modeling. Stage 3 applies neural style transfer trained on real seismic to produce realistic textures while preserving fault labels throughout.

Choi et al., 2025).

To bridge this gap, we introduce **SeisMark**, a standardized benchmark designed to strictly validate 3D seismic interpretation models. We resolve the trade-off between label accuracy and textural realism by employing a novel hybrid pipeline. We construct diverse geological structures (folds and faults) alongside a generative texture synthesis module to simulate realistic field characteristics. Rather than relying solely on physics-based approximations, our approach performs a stochastic style transfer, leveraging diffusion-based refinement to inject the chaotic noise and complex spectral properties of real-world surveys into our synthetic volumes. This process strictly enforces a label-preserving constraint, ensuring that the resulting photorealistic textures align perfectly with the underlying structural fault labels.

Our contributions can be summarized as follows:

1. **Roadmap for Evaluation:** We identify the lack of reliable ground-truth benchmarks for 3D seismic interpretation and formally define the problem as a 3D segmentation task under significant domain shift.
2. **The SeisMark Benchmark:** We release a first-of-its-kind open benchmark featuring a realistic synthetic 3D seismic volumes ($1500 \times 1500 \times 1200$) with verified fault labels. To ensure validity, we employ a hybrid curation pipeline that injects realistic field textures while strictly preserving the structural integrity of physics-based ground truth, enabling safe evaluation for critical applications like CCS.
3. **Comprehensive Evaluation:** We evaluate distinct generations of domain-specific segmentation architectures (e.g., FaultSeg3D V1 (Wu et al., 2019) vs. V2 (Li et al., 2024)) on SeisMark. This comparative analysis quantifies the impact of realistic texture on model performance, establishing a rigorous baseline for future research.

## 2. Related Work

**Physics-Based Synthetic Data in Geophysics.** To overcome the label scarcity and subjective bias inherent in real-world surveys, the field has relied heavily on synthetic data. The SEG Advanced Modeling (SEAM) corporation established industry-standard benchmarks using rigorous wave-equation modeling (Fehler & Keliher, 2011). However, producing these models and simulations demands substantial organizational resources and multi-year development cycles, which limits scalability and confines the dataset to a handful of curated geological scenarios rather than the thousands of structural variations needed for a truly general benchmark. In the academic sphere, the FaultSeg3D dataset (Wu et al., 2019) democratized access utilizing simplified 1D convolutional modeling. While efficient, these approximations yield overly clean textures that lack the diffraction patterns, multiples, and non-stationary noise of field records. Consequently, these datasets serve as inadequate benchmarks; they present an overly optimistic evaluation landscape where models can achieve near-perfect metrics without possessing the robustness required for real-world inference.

**Benchmarks in Subsurface Learning.** The integration of machine learning into subsurface modeling has been accelerated by the arrival of standardized benchmarks. Open-FWI (Deng et al., 2022) serves as a defining proof-of-concept, demonstrating how open-source synthetic datasets can democratize research in the subsurface domain. However, OpenFWI is constrained by both its task and its spatial scope. It is specifically designed for Full Waveform Inversion (FWI)—a regression task focused on estimating physical velocity parameters—and relies on small-scale image cubes that do not reflect the complexity of large, field-scale volumes. Consequently, it does not address the downstream needs of seismic interpretation, which operates

on migrated reflectivity seismic volumes. Currently, there is no interpretation-focused equivalent that matches the scale and impact of OpenFWI, leaving a critical void for tasks like fault detection that rely on high-frequency, field-scale structural analysis.

**Generative Models for Domain Adaptation.** To bridge the "Sim2Real" gap without sacrificing label precision, generative domain adaptation has seen increasing adoption. Early approaches utilized Neural Style Transfer (NST) (Gatys et al., 2015) or CycleGANs (Zhu et al., 2017) to superimpose realistic textures onto synthetic fault volumes (Jing et al., 2019; Ferreira et al., 2019). However, adversarial methods are prone to hallucinations—generating realistic-looking textures that unintentionally alter or destroy the underlying geological structures (e.g., creating fake faults or erasing real ones). More recently, Diffusion Probabilistic Models have emerged as a robust alternative for controlled synthesis (Sohl-Dickstein et al., 2015; Ho et al., 2020; Song et al., 2021; Dhariwal & Nichol, 2021). This shift parallels recent advancements in general computer vision, where diffusion-based style transfer has successfully bridged the domain gap between simulated driving scenarios and real-world imagery for segmentation tasks (Chigot et al., 2025). Our work extends these principles to the geophysical domain, employing a physics-constrained generative pipeline to inject realistic aleatoric noise while strictly enforcing structural consistency with the ground truth labels.

## 3. Dataset Curation

Benchmarking 3D seismic fault segmentation is limited by the lack of scalable, verifiable ground truth: field labels are sparse and interpreter-dependent, while purely physics-based synthetic volumes are typically too clean to reflect real migrated seismic texture. This motivates the following research question:

> *"Can we generate 3D seismic volumes that are simultaneously (a) label-accurate by construction and (b) texturally realistic enough to stress-test modern segmentation models under domain shift?"*

To make this question actionable, we translate it into three benchmark design goals:

- **Geological diversity:** capture a broad range of stratigraphic and structural configurations beyond a single curated scenario;
- **Physics-consistent structure:** ensure reflector geometry and fault offsets are consistent with a forward imaging model;
- **Label-preserving realism:** inject field-like texture without creating, removing, or shifting faults relative to the ground-truth labels.

Our approach, **SeisMark** meets these requirements with a hybrid pipeline that decouples *structure* from *texture*.

Specifically, our synthetic data generation pipeline comprises three components: (1) a **procedural model builder** that constructs geologically plausible 3D velocity models with precise fault labels, (2) a **forward modeling engine** that converts these velocity models into pseudo-seismic volumes, and (3) a **diffusion-based style transfer** module that injects field-like texture while preserving the labeled structures (Figure 1).

### 3.1. Procedural Model Building

Our **procedural model builder** employs an event-based modular architecture where each geological process is represented by a parameterized module. The core container maintains three-dimensional arrays for P-wave velocity, layer identifiers, and fault labels. To ensure structural realism, we base our simulation on the Gorgon project, a well-characterized gas field and $CO_2$ sequestration site in the Northern Carnarvon Basin of Western Australia that exhibits the full complexity of a polyphase extensional basin: primary horst-bordering faults, subordinate synthetic faults, polygonal fault tiers, and prograding carbonate clinoforms (McCormack & McClay, 2013).

Each module in **model builder** uses mathematical operations that mirror real geological mechanisms. The **Deposit** module creates sedimentary layers with 3D simplex noise for sub-seismic velocity heterogeneity in space, randomized interbed boundary capturing the gradational contacts observed in real stratigraphy. The **Squish** module applies differential compaction using 2D fractal noise fields rotated to match structural trends, producing realistic thickness variations and gentle folding (Allen & Allen, 2013). The **Fault** module implements listric normal fault geometry through coordinate rotation (aligning the fault plane with strike direction) and depth-varying dip angles that flatten toward a detachment surface—matching the geometry of extensional faults in rifted basins (Gibbs, 1984). The **Stratigraphic** module generates clinoform geometries and models cyclic sequences of progradation, aggradation, and retrogradation (Posamentier & Vail, 1988) that replicate the Paleogene/Neogene carbonate shelf margin patterns observed in the Gorgon region.

The model is constructed layer-by-layer following geological superposition (Figure 2). Initial basement and sedimentary deposits (Deposit module) undergo differential compaction (Squish module), creating gentle antiforms and layer thickness variations. Pre-fault layers are then deposited, followed by fault emplacement: 5 primary listric normal faults with NNE-trending azimuths matching dominant Gorgon fault families, plus 488 subsidiary synthetic and antithetic faults distributed around the primary fault systems. Post-fault inclined deposits model the Upper Cretaceous Gearle Siltstone, prograding carbonate clinoforms (Stratigraphic module) represent Neogene sequences, and finally an ero-

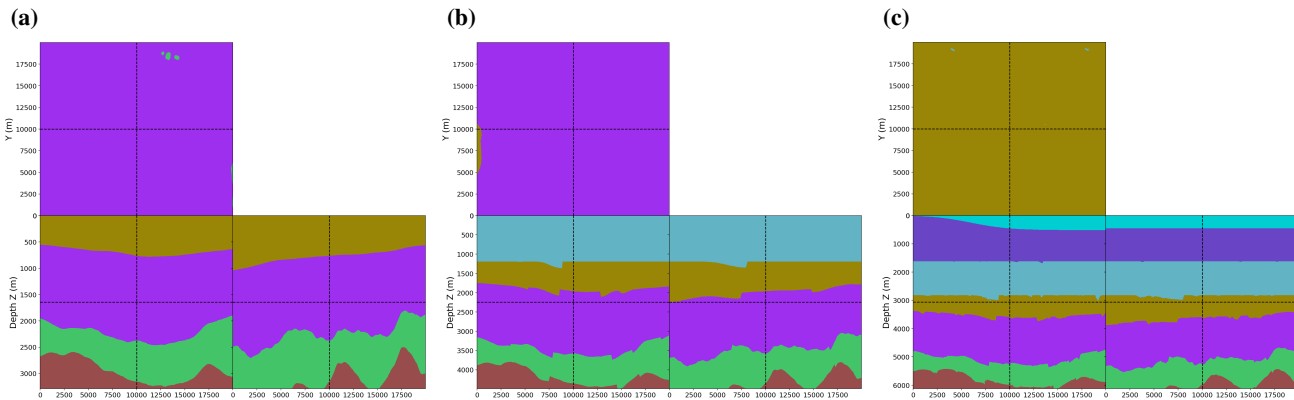

*Figure 2.* **Procedural model evolution.** Orthogonal slices show uniquely colored stratigraphic units. **(a)** Initial state: Four sedimentary deposits exhibiting Perlin noise heterogeneity and differential compaction. **(b)** Intermediate state: Addition of 5 primary listric faults and a thick Cretaceous deposit, resulting in syn-depositional deformation with gentle folding. Subsequently, 488 subsidiary faults are injected. **(c)** Final model (14 layers): Includes post-fault siltstone drape, six prograding carbonate clinoform cycles, erosional truncation, and water column fill. Note: The final volume used for fault detection was cropped to a depth of 6000 m.

sion surface with fractal bathymetric variation creates the modern seafloor. See Appendix A for detailed formation mapping and Appendix B for module implementations.

### 3.2. Forward Modeling Engine: Pseudo-Seismic Generation

The **forward modeling engine** converts the labeled velocity models into pseudo-seismic volumes using a 1D convolutional approach. We first calculate vertical reflectivity series from the impedance contrasts derived from the velocity model (using Gardner's relation for density (Gardner et al., 1974)). These reflectivity spikes are then convolved with a zero-phase Ricker wavelet (central frequency 25 Hz) to simulate the band-limited nature of seismic data.

While 1D convolution approximates a post-stack time-migrated volume, it simplifies complex wave-propagation effects. We intentionally employ this lightweight structural proxy, delegating the simulation of complex wave physics (such as diffractions, multiples, and Fresnel zone smoothing) to the subsequent generative diffusion stage. This design choice enables the rapid generation of thousands of volumes without the computational bottleneck of full-wavefield modeling, while ensuring the final texture is driven by data-learned priors rather than approximate physical equations.

### 3.3. Bridging the Sim2Real Gap: Realistic Texture Synthesis

While the pseudo-seismic volume $S_{\text{pseudo}}$ faithfully reproduces fault positions and layer geometries and includes bed-scale, laterally coherent amplitude variation from procedural noise/interbed randomization, it lacks the non-stationary heterogeneous texture characteristic of real migrated seismic volumes. To bridge this gap without sacrificing label precision, we employ a diffusion-based image-to-image

translation module.

We cast texture injection as a *conditional refinement* task. To strictly decouple structural geometry from textural noise, we employ a diffusion-based image-to-image translation framework. The pseudo-seismic volume serves as a strong structural condition (control signal), locking the phase and position of major reflectors. The diffusion model, trained on field data (F3 or Gorgon), iteratively denoises the volume starting from a state of partial noise.

**Label Preservation & Texture Intensity.** The label-preserving constraint is enforced via the conditioning mechanism: the model is penalized for deviations from the structural edges defined in the input pseudo-seismic. We control the Texture Intensity (Low vs. High) by modulating the initial noise level (timestep $t$ in the forward diffusion process) and the guidance scale. A lower intensity modifies only high-frequency spectral content (fine texture), while higher intensity allows for modification of mid-frequency amplitudes, simulating stronger acquisition artifacts while retaining the macro-structural skeleton defined by the ground truth.

**Why diffusion and not adversarial domain adaptation?** Earlier work on GAN-based seismic synthesis (Ferreira et al., 2019) (and unpaired image-to-image translation more broadly (Zhu et al., 2017)) optimizes a distributional objective with no explicit per-voxel structural constraint, and is prone to mode collapse and to hallucinating textures that destroy or fabricate the very fault discontinuities the benchmark must preserve. We instead choose a diffusion formulation precisely because the partial-noising / reverse-denoising procedure begins from a partially noisy pseudo-seismic state rather than pure random noise: the macroscopic geometry, fault offsets, and main reflector phases stay locked in place while only high-frequency texture is regenerated.

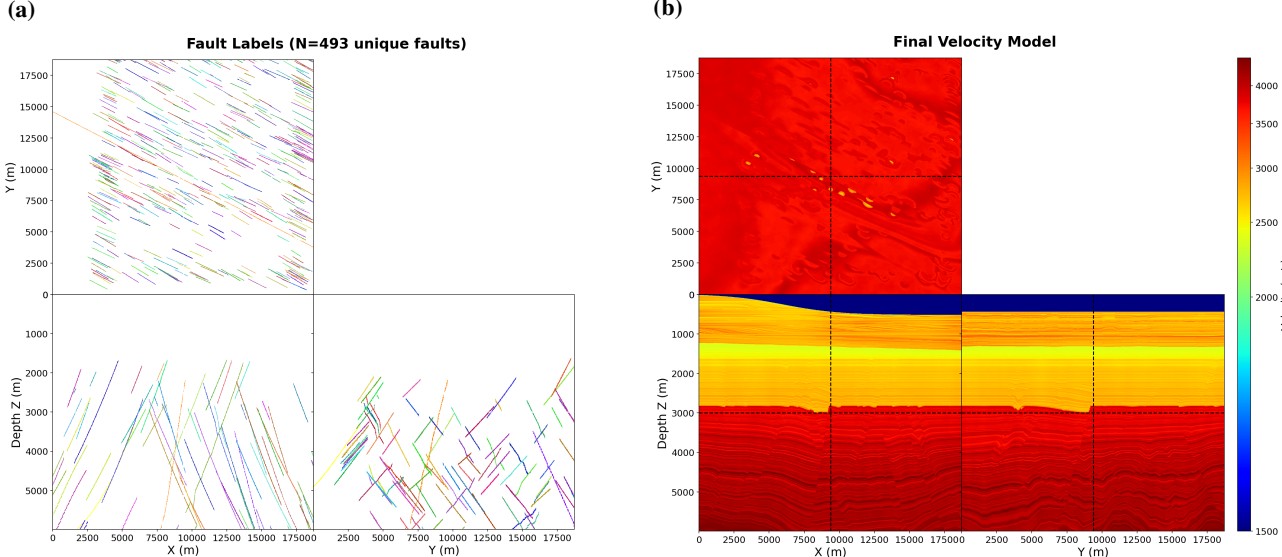

**Figure 3.** Faults network and final geological velocity model. **(a)** Fault labels with each of the 493 faults uniquely colored: 5 primary NNE-trending listric normal faults (large continuous structures) plus 488 subsidiary synthetic and antithetic faults distributed around the primary systems. **(b)** Final P-wave velocity model showing the complete geological structure. In this viewpoint, velocity ranges from 1500 m/s (water column, blue) through 2000–2600 m/s (post-rift sediments) to ∼4200 m/s (Late Triassic, red).

**The Two-Component Workflow.** Our hybrid approach deliberately separates concerns: the model builder provides ground-truth structural labels and physically consistent geometry (the "skeleton"), while Diffusion Styling adds realistic texture while preserving structural alignment (the "skin"). The result is synthetic seismic that is simultaneously label-accurate for training supervision and texturally realistic for domain transfer to real data.

**Basin-Specific Texture Transfer.** The visual character of migrated seismic volumes varies substantially with acquisition geometry, processing workflow, and target depth—factors that collectively shape the "texture" overlaid on sub-surface structure. To capture this diversity, we train separate diffusion models on two datasets representing contrasting acquisition eras and geometries, enabling domain-specific styling at two intensity levels (low and high).

The **Netherlands F3** survey is a 3D volume acquired in 1987 using narrow-azimuth towed streamers in the Dutch North Sea (Alaudah et al., 2019). The post-stack time-migrated cube spans approximately 384 km$^2$ with 25 m bin spacing and images relatively shallow stratigraphy (0–1.85 s two-way time, corresponding to ∼1.2 km depth). The data exhibit characteristic vintage streamer texture: visible acquisition footprint aligned with sail-line geometry and amplitude variations across the volume (Ishak et al., 2018). Reflector continuity is generally good in the upper clino-form packages, with the texture reflecting the single-azimuth acquisition and post-stack processing typical of that era.

The **Gorgon** survey, by contrast, is a 2015–2016 full-azimuth ocean-bottom-node (OBN) acquisition over the Gorgon gas field in the Northern Carnarvon Basin, off-shore Western Australia (Chambath et al., 2019). Approximately 3,100 seafloor nodes recorded a dense shot carpet (∼18.75 m inline shot spacing) yielding high fold (∼289) and full-azimuth illumination to depths exceeding 7 km (Scholtz et al., 2023; Operto et al., 2023). The resulting migrated cube exhibits broadband frequency content, minimal acquisition footprint, and continuous reflectors even in structurally complex fault-shadow zones (Chambath et al., 2019). The characteristic texture of OBN data reflects residual effects of clock drift, node positioning, and water-velocity corrections applied during processing (Scholtz et al., 2023).

These two datasets provide complementary texture priors representing different acquisition technologies and depth regimes (Figure 4). By offering both styling options, Seis-Mark enables users to generate synthetic training data whose visual statistics match the characteristics of their target interpretation domain—whether vintage archive data or contemporary broadband surveys.

Figure 1 summarizes the SeisMark generation pipeline. Given a random seed and a set of geological parameters, the **procedural model builder** produces a 3D P-wave velocity model $V \in \mathbb{R}^{D \times H \times W}$ together with voxel-aligned fault labels $Y \in \{0,1\}^{D \times H \times W}$. Next, a **forward modeling engine** converts $V$ into a pseudo-seismic volume $S_{\text{pseudo}} \in \mathbb{R}^{D \times H \times W}$ that preserves reflector geometry and fault offsets implied by the underlying velocity structure. Finally, a **diffusion-based style transfer** module transforms $S_{\text{pseudo}}$ into a styled volume $S_{\text{styled}}$ whose texture matches the statistics of real migrated seismic, while maintaining

**(a)**

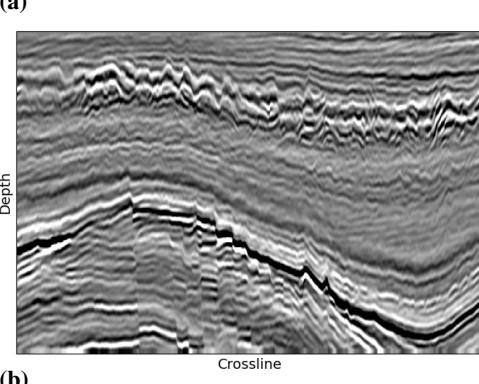

**(b)**

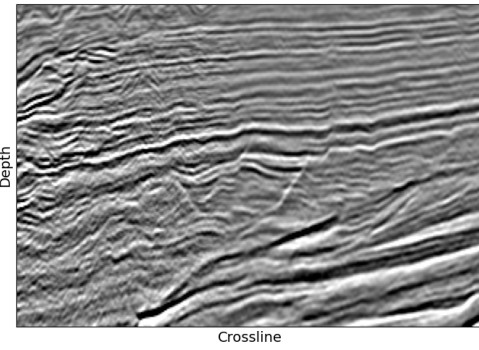

*Figure 4.* Texture comparison of 2D migrated seismic patches from the **(a)** Netherlands F3 and **(b)** Gorgon datasets. Both patches were extracted from comparable shallow depths to highlight textural differences arising from acquisition and processing rather than structural geology.

alignment with the ground-truth labels $Y$. The result is a set of texture-enhanced synthetic volumes $S_{\text{styled}}$ that retain the label accuracy and structural diversity of procedural modeling, while more closely matching the visual statistics of real migrated seismic—thereby bridging the Sim2Real gap for benchmark evaluation.

## 4. The SeisMark Benchmark

Applying the pipeline described in Section 3, we constructed **SeisMark**, a large-scale standardized benchmark for 3D seismic interpretation. This section details the dataset specifications, available variants, and statistical characteristics. All datasets will be made publicly available upon acceptance of the paper.

### 4.1. Dataset Specifications

The benchmark consists of a single contiguous volume spanning $18.75 \times 18.75 \times 6.0$ km ($1500 \times 1500 \times 1200$ voxels). Unlike patch-based datasets (e.g., OpenFWI), this deliberate field-scale sizing preserves long-range geological continuity. The volume features a fully annotated, bimodal network of 493 faults. This comprises 5 **Primary Faults**: large-scale, NNE-trending listric normal faults with significant displacement ($> 100$m) penetrating the entire stratigraphic column, and 488 **Subsidiary Faults**: a dense cloud of smaller synthetic and antithetic fractures. With subtle offsets ($< 20$m)

and complex intersection geometries (Y- and X-junctions), these subsidiary faults capture the "long tail" of interpretation challenges often missed by current auto-trackers.

### 4.2. Benchmark Variants

To support controlled robustness testing, we release the dataset in five distinct variants (Table 1). All variants share the exact same voxel-wise ground truth labels $Y$, allowing for direct localized performance comparison.

We provide a **Clean (Baseline)** variant, which is the direct output of the forward modeling engine. This represents the ideal physics scenario common in existing synthetic datasets, characterized by high signal-to-noise ratio and perfect reflector continuity. Alongside this, we provide four **Styled** variants generated via our diffusion pipeline, stratified by **Source Domain** (North Sea vs. Australian Basin) and **Texture Intensity** (Low vs. High). Because the pipeline decouples procedural structural generation (the "skeleton") from diffusion-based style transfer (the "skin"), every styled variant shares an *identical* voxel-wise ground-truth label set with the Clean baseline. To our knowledge, no prior synthetic seismic benchmark exposes the texture axis independently of the structural axis at this scale, making SeisMark the first benchmark capable of explicitly testing a fault detector's robustness to acoustic domain shifts while holding the geology fixed. This structure enables a ladder of difficulty: models can first be validated on Clean data to verify geometrical understanding, and then progressively stress-tested on Styled variants to measure robustness to domain shift.

*Table 1.* Benchmark composition. The total release comprises $\sim 65$ GB of volumetric data, providing a comprehensive testbed for 3D segmentation models. All variants share identical ground truth.

| Variant Name | Texture Source | Intensity | Size (GB) |
|---|---|---|---|
| SeisMark-Clean | Physics-only | N/A | 10.8 |
| SeisMark-F3-L | North Sea | Low | 10.8 |
| SeisMark-F3-H | North Sea | High | 10.8 |
| SeisMark-Aus-L | Australia | Low | 10.8 |
| SeisMark-Aus-H | Australia | High | 10.8 |
| **Ground Truth** | Fault Labels | — | 10.8 |

## 5. Experiments

We design our experiments to address two fundamental questions regarding the SeisMark benchmark: *Validity*—does the texture synthesis pipeline preserve the underlying geological structure? and *Utility*—does the injected realism effectively expose the brittleness of current state-of-the-art models?

*Table 2.* **Comprehensive Benchmark Results.** Four pre-trained fault detectors (FaultSeg3D (Wu et al., 2019), FaultSeg3D+ (Li et al., 2024), Fault-Net (Dou et al., 2022), RGF (Lin et al., 2025)) evaluated zero-shot across the five SeisMark variants. Numbers are extracted programmatically from our benchmark output. At strict tolerance ($T = 0$) all models score low due to the ambiguity of sub-voxel fault localization within band-limited wavelets; at valid geological tolerances ($T \geq 1$), SeisMark produces a clear discriminative gradient. Fault-Net (trained with real-data priors) leads on every variant, while RGF and the legacy V1 collapse under high-intensity textures.

| Variant | Model | IoU ↑ | Tolerance = 0 | | | Tolerance = 1 | | | Tolerance = 2 | | |
|---|---|---|---|---|---|---|---|---|---|---|---|
| | | | P | R | F1↑ | P | R | F1↑ | P | R | F1↑ |
| **Clean (Baseline)** | FaultSeg3D | 0.262 | 0.313 | 0.614 | 0.415 | 0.417 | 0.804 | 0.549 | 0.435 | 0.864 | 0.579 |
| | FaultSeg3D+ | 0.304 | 0.634 | 0.368 | 0.466 | 0.676 | 0.792 | 0.730 | 0.687 | 0.855 | 0.762 |
| | Fault-Net | **0.441** | 0.652 | 0.576 | **0.612** | 0.805 | 0.790 | **0.798** | 0.824 | 0.847 | **0.836** |
| | RGF | 0.342 | 0.506 | 0.512 | 0.509 | 0.633 | 0.700 | 0.665 | 0.643 | 0.767 | 0.700 |
| **North Sea (Low)** | FaultSeg3D | 0.258 | 0.520 | 0.338 | 0.410 | 0.709 | 0.489 | 0.579 | 0.748 | 0.567 | 0.645 |
| | FaultSeg3D+ | 0.272 | 0.807 | 0.290 | 0.427 | 0.902 | 0.628 | 0.740 | 0.913 | 0.706 | **0.796** |
| | Fault-Net | **0.389** | 0.642 | 0.497 | **0.560** | 0.868 | 0.646 | **0.741** | 0.900 | 0.711 | 0.794 |
| | RGF | 0.238 | 0.524 | 0.304 | 0.385 | 0.768 | 0.450 | 0.567 | 0.820 | 0.536 | 0.648 |
| **North Sea (High)** | FaultSeg3D | 0.167 | 0.385 | 0.228 | 0.287 | 0.560 | 0.358 | 0.437 | 0.614 | 0.442 | 0.514 |
| | FaultSeg3D+ | 0.203 | 0.754 | 0.218 | 0.338 | 0.884 | 0.494 | 0.633 | 0.900 | 0.579 | **0.705** |
| | Fault-Net | **0.297** | 0.593 | 0.373 | **0.458** | 0.831 | 0.512 | **0.634** | 0.875 | 0.581 | 0.699 |
| | RGF | 0.151 | 0.396 | 0.196 | 0.262 | 0.615 | 0.322 | 0.423 | 0.687 | 0.417 | 0.519 |
| **Australia (Low)** | FaultSeg3D | 0.249 | 0.542 | 0.315 | 0.399 | 0.746 | 0.457 | 0.567 | 0.783 | 0.529 | 0.632 |
| | FaultSeg3D+ | 0.250 | 0.808 | 0.266 | 0.401 | 0.921 | 0.578 | 0.711 | 0.931 | 0.657 | 0.771 |
| | Fault-Net | **0.384** | 0.645 | 0.488 | **0.555** | 0.890 | 0.625 | **0.735** | 0.921 | 0.685 | **0.786** |
| | RGF | 0.237 | 0.527 | 0.301 | 0.384 | 0.787 | 0.443 | 0.567 | 0.843 | 0.525 | 0.647 |
| **Australia (High)** | FaultSeg3D | 0.145 | 0.388 | 0.188 | 0.253 | 0.578 | 0.305 | 0.400 | 0.638 | 0.385 | 0.480 |
| | FaultSeg3D+ | 0.154 | 0.725 | 0.164 | 0.267 | 0.889 | 0.388 | 0.540 | 0.908 | 0.471 | 0.620 |
| | Fault-Net | **0.278** | 0.571 | 0.352 | **0.435** | 0.834 | 0.480 | **0.609** | 0.886 | 0.545 | **0.675** |
| | RGF | 0.143 | 0.389 | 0.185 | 0.250 | 0.621 | 0.306 | 0.410 | 0.706 | 0.399 | 0.510 |

## 5.1. Dataset Validation: Structural Fidelity

A critical requirement for a synthetic benchmark is label integrity. Since our pipeline injects heavy textural noise into the pseudo-seismic volumes, we must verify that this process does not hallucinate new edges or displace existing fault discontinuities relative to the ground-truth labels. We employ reference-based image quality assessment metrics to quantify structural consistency between the input (Pseudo-Seismic) and output (Styled) volumes.

We utilize the Structural Similarity Index Measure (SSIM) (Wang et al., 2004) to measure the preservation of macro-scale geological features. However, global metrics can be insensitive to local edge shifts. To specifically validate the stability of fault discontinuities, we compute the Gradient Magnitude Correlation (GMC) (Xue et al., 2013), defined as the Pearson correlation coefficient between the gradient magnitude maps of the pseudo-seismic and styled volumes. Since seismic faults are defined by sharp local gradients, a high GMC score ($> 0.87$) confirms that the edges (fault locations) have not been displaced or hallucinated during the styling process. Both SSIM and GMC are computed over the entire seismic cube to ensure a holistic assessment of structural fidelity.

As shown in Table 3, the pipeline successfully decouples texture generation from structural integrity. Across all vari-

*Table 3.* **Structural Fidelity Analysis.** High SSIM scores ($> 0.89$) confirm that the global geological structure is preserved across all variants. The variation in Gradient Correlation (GMC) quantifies the intensity of the injected noise without indicating structural shift.

| Style Source | Intensity | SSIM ↑ | GMC ↑ |
|---|---|---|---|
| North Sea (F3) | Low | 0.92 | 0.91 |
| | High | 0.88 | 0.87 |
| Australia (Gorgon) | Low | 0.97 | 0.93 |
| | High | 0.96 | 0.90 |

ants, SSIM scores remain consistently high (0.88–0.97), indicating that the bulk geological geometry—layer dips and folding patterns—is preserved. Furthermore, the Gradient Magnitude Correlation remains robust ($> 0.87$) even in High Intensity settings. While lower than the clean input, this value represents a valid alignment: the correlation reduces not because faults have moved, but because the styling introduces additional high-frequency gradients (simulated noise) that do not exist in the clean input.

## 5.2. Baseline Evaluation: Stress-Testing Robustness

We evaluate four pre-trained fault segmentation networks. **FaultSeg3D (V1)** (Wu et al., 2019) is a standard U-Net trained on synthetic data with simple random noise. **FaultSeg3D+ (V2)** (Li et al., 2024) is a deeper residual network

| FaultSeg3D (V1) | FaultSeg3D+ (V2) | Fault-Net | RGF |
| --- | --- | --- | --- |

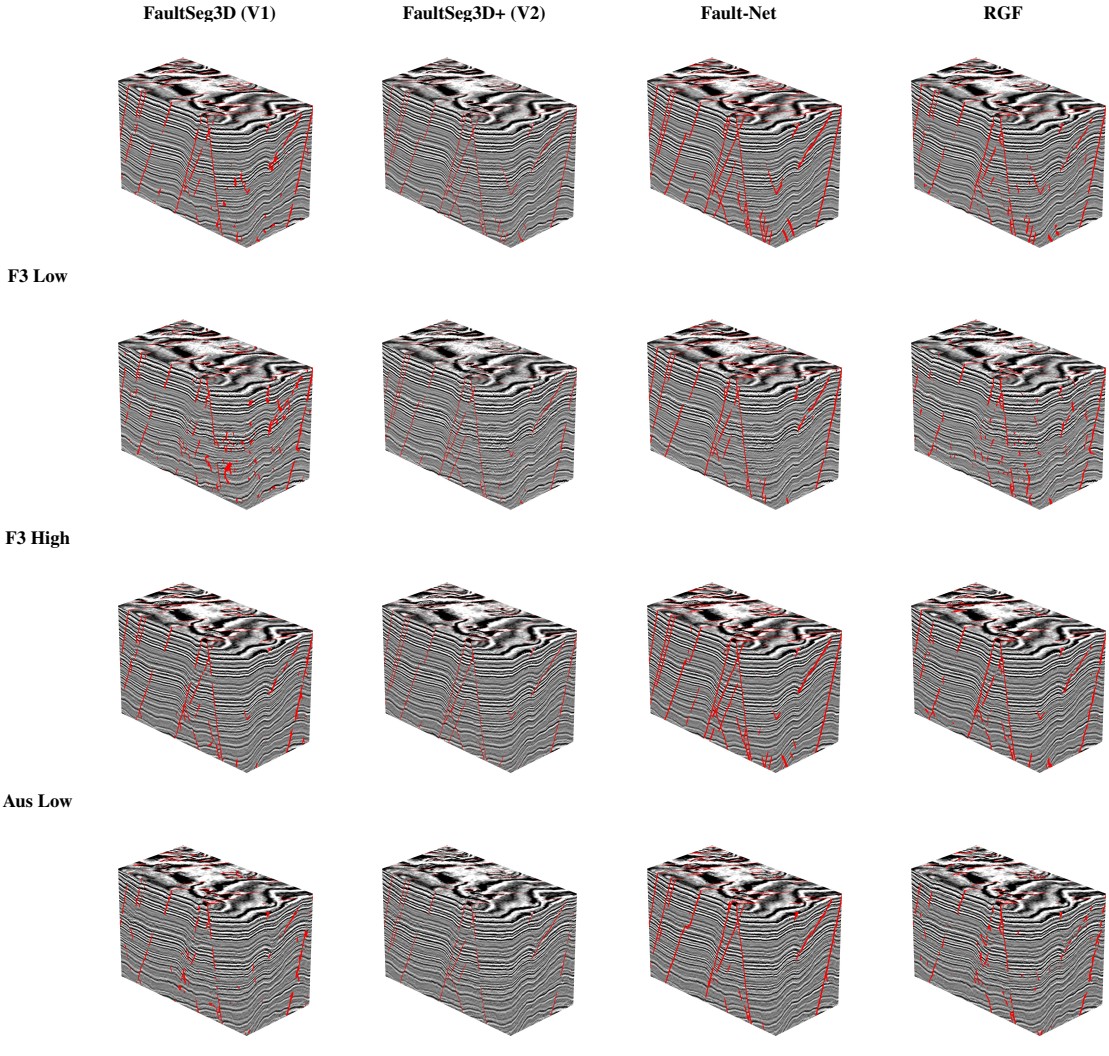

*Figure 5.* **Qualitative stress-test on SeisMark.** 3D sub-cube renderings of predicted fault probabilities (red) overlaid on the styled seismic for each pre-trained model (columns) on each styled SeisMark variant (rows). The legacy FaultSeg3D (V1) and RGF detectors visibly sparsify their predictions on high-intensity variants, while Fault-Net and FaultSeg3D+ (V2) retain coherent fault planes, mirroring the IoU/F1 gradient in Table 2.

trained with a curriculum of more realistic noise. **Fault-Net** (Dou et al., 2022) uses a MultiScale Compression Fusion block to keep thin edge-like fault features intact during feature propagation, and is trained on a mixture of synthetic data and sparse manually-labeled 2D slices from real 3D field surveys. **RGF** (Lin et al., 2025) is a 3D U-Net trained on a broader synthetic dataset that includes dips, folds, unconformities, channels, caves, salt bodies, and composite faults.

**Experimental Setup.** We adopt a standard Sim2Real evaluation protocol to ensure reproducibility. Both the V1 and V2 models utilize **pre-trained weights from the original publications**, and are evaluated zero-shot on the Styled variants. This rigorously tests robustness to domain shift. We report Intersection-over-Union (IoU) and standard segmentation metrics (Precision, Recall, F1).

To account for the inherent structural uncertainty of thin fault lines, we define true positives using a spatial tolerance distance $T$. Specifically, we apply 3D morphological binary dilation (using a $3 \times 3 \times 3$ connectivity structure, iterated $T$ times). For Recall, we dilate the prediction mask before intersecting with the ground truth; for Precision, we dilate the ground truth mask before intersecting with the prediction. We report results for $T \in \{0, 1, 2\}$.

**Results and Analysis.** The quantitative results (Table 2) and qualitative visualizations (Figure 5) reveal four critical findings that validate SeisMark's utility:

1. **Recall Saturation on Simplified Data:** On the Clean baseline, V1, V2, and Fault-Net all achieve similarly high detection rates (Recall $\approx$ 0.79–0.81 at $T = 1$). This *recall saturation* suggests that simplified physics-based data is insufficiently complex to test a model's

sensitivity. F1 scores remain depressed on Clean data due to low precision caused by physics-modeling artifacts that mimic faults.

2. **Wide Score Spread Across Models:** On the styled variants, IoU scores differ sharply between models. On Australia (Low), Fault-Net scores IoU 0.384 and RGF scores 0.237, a gap of 0.147 IoU. Fault-Net leads in IoU on all five variants. V1 also degrades on every styled variant compared to its Clean baseline (Recall at $T = 1$: Clean 0.804, F3-Low 0.489, Aus-High 0.305), with the largest drop on High Intensity textures, while V2 and Fault-Net stay more stable. We attribute Fault-Net's lead to its real-world labeled training slices; the other three models train only on synthetic data, which is the asymmetry the styled variants are built to expose.

3. **Precision–Recall Tradeoff:** V2 and Fault-Net trade Precision against Recall in different ways on the styled variants. V2 pushes its Precision high (P@$T = 1$ reaches 0.921 on Australia Low) but its Recall falls to 0.578 on the same variant, a $|P - R|$ gap of 0.343. Fault-Net keeps the two close: on the Clean baseline its gap is 0.015 ($P = 0.805$, $R = 0.790$), and on Australia Low its gap is 0.265, about 23% tighter than V2's gap on the same variant. A model that is useful in the field needs to keep both Precision and Recall up at the same time, not push one metric while the other drops, so this balance matters more than peak Precision on a single variant.

4. **Diffusion Styling Removes Fake-Fault Artifacts:** Two different detectors get *higher* Precision on Low Intensity styled variants than on the Clean baseline. V2's P@$T = 1$ rises from 0.676 on Clean to 0.902 on F3-Low and 0.921 on Aus-Low. Fault-Net's P@$T = 1$ rises from 0.805 to 0.868 and 0.890. Two different models moving the same direction is a sign the cause is in the data, not the model. The forward modeling stage creates sharp interference patterns that look like faults to a detector and trigger false positives on the Clean cube (Figure 5); the diffusion styling acts as a coherence filter, healing these artifacts into naturally continuous reflectors so that the same detectors trigger fewer false positives on the styled volumes.

## 6. Conclusions

In this work, we introduced **SeisMark**, a large-scale, open-source benchmark designed to resolve the long-standing data paradox in seismic interpretation. For decades, the geophysical community has been forced to choose between the structural certainty of simplified synthetic data and the textural realism of unlabelled field records. SeisMark bridges this divide by integrating procedural geological modeling with diffusion-based texture synthesis. Our contributions are threefold. First, we released a 2.7-billion-voxel volume that captures the geological complexity of real-world basins, labeled with pixel-perfect precision. Second, we validated the structural integrity of this dataset using SSIM and GMC metrics, confirming that our styling pipeline preserves label accuracy while injecting realistic domain textures. Finally, our benchmarking experiments demonstrated that SeisMark is essential for evaluating deployment readiness. We showed that on simplified synthetic data, legacy models trained on random noise achieve detection rates indistinguishable from state-of-the-art models. On the styled variants, that apparent parity disappears: legacy models suffer a performance collapse, while modern architectures stay closer to their Clean-baseline scores. This discriminative capability establishes SeisMark as a necessary standard for the next generation of AI geophysics. Future extensions include masked SSIM and gradient correlation computed strictly within morphologically dilated fault zones, additional procedural volumes covering compressional and salt-dominated basins, and a direct head-to-head numerical comparison against Cycle-GAN/NST adversarial styling pipelines.

## Impact Statement

This work contributes to the development of reliable machine learning systems for subsurface characterization, a domain critical to the global energy transition. The primary societal impact of SeisMark lies in its potential to enhance the safety and efficacy of Carbon Capture and Storage (CCS) and geothermal energy projects. Accurate fault detection is a prerequisite for securing gigaton-scale storage; undetected faults can act as "thief zones" or leakage pathways for injected $CO_2$, compromising long-term seal integrity and environmental safety. By providing a rigorous benchmark to stress-test interpretation models, this research aims to mitigate the risks of deployment failures in these high-stakes environments.

Furthermore, this work addresses the ethical and technical challenges of applying generative AI to scientific domains. While diffusion models offer realistic texture synthesis, they carry the risk of hallucinating non-existent geological features. Our pipeline explicitly mitigates this by enforcing strict structural conditioning and validating label integrity via quantitative metrics (SSIM, GMC), ensuring that the synthetic data remains physically consistent. Finally, by releasing an open-source, field-scale benchmark, we aim to democratize access to geophysical research, allowing the broader machine learning community to contribute to energy-transition solutions previously restricted by the scarcity of proprietary industry data.

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

# A. Geological Foundation

## A.1. The Gorgon Field Analog

To ensure our synthetic models capture the structural complexity of real subsurface environments, we base our simulation on the Gorgon project, a well-characterized gas field located approximately 130 km offshore in the Northern Carnarvon Basin of Western Australia (McCormack & McClay, 2013). Notably, the Gorgon Project also hosts one of the world's largest $CO_2$ sequestration operations, making this structural analog relevant for both hydrocarbon and carbon storage applications. The Gorgon field forms a NNE-trending horst block at the southwestern end of the Rankin Trend within the Westralian Superbasin (Longley et al., 2002), bounded by extensional faults that developed during polyphase Late Triassic to Early Cretaceous rifting associated with Gondwana breakup.

## A.2. Stratigraphic Succession

The stratigraphic succession spans approximately 300 million years, from Permian basement through Neogene carbonates (Hocking, 1988):

*Table 4.* Stratigraphic succession and module mapping.

| Age | Formation | Play Int. | Velocity | Module | Seismic Character |
|---|---|---|---|---|---|
| Permian | Locker Shale, Kennedy Gp | PZ50 | ~4800 m/s | Continental shelf | Low-amp., semi-cont., NW-dip |
| Triassic | Mungaroo Fm | TR10-TR20 | 3800–4200 m/s | Deposit + Squish | High-amp., cont., gently folded |
| Late Triassic | Brigadier Fm | TR30 | 3200–3600 m/s | Deposit | Trans. fluvial-deltaic to marine |
| Jurassic | Murat Silt., Dingo Clay. | J20-J50 | 2800–3200 m/s | Deposit (siltstone) | Low-amp., discont., fault-def. |
| Early Cretaceous | Flacourt Fm (Barrow Gp) | K10-K20 | 2400–2600 m/s | Inclined Deposit | Shallow-dip, NNW-prog. foresets |
| Lower-Upper Cret. | Muderong Sh., Gearle Silt. | K30-K50 | 2200–2400 m/s | Deposit (parallel) | Low-amp., cont. parallel refl. |
| Paleogene | Cardabia Calc., Giralia Calc. | T10-T20 | 2100–2200 m/s | Stratigraphic | Passive margin carb. progr. |
| Neogene | Cape Range Gp, Trealla Lst. | T30 | 2000–2100 m/s | Stratigraphic | High-amp., NW-prog. clinoforms |
| Present | Water Column | — | 1500 m/s | Erosion | Modern seafloor bathymetry |

The primary reservoir interval is the Triassic Mungaroo Formation (TR20), comprising fluvio-deltaic meander belt sandstones up to 50 m thick interbedded with overbank siltstones and shales. The Lower Cretaceous Muderong Shale (K30) provides the regional top-seal, marking the onset of post-rift thermal subsidence. The overlying Upper Cretaceous Gearle Siltstone (K50) and Paleogene-Neogene carbonates represent quiescent drift-stage deposition characterized by low-amplitude, parallel, continuous reflectors—which we replicate using our Deposit module with minimal velocity contrast and gentle layer-parallel geometry. The Stratigraphic module generates the high-amplitude sigmoidal clinoform geometries characteristic of the Neogene carbonate shelf margin progradation (Longley et al., 2002).

## A.3. Fault Populations

The Gorgon field exhibits distinct fault populations that we replicate: **(1) Primary faults**—NNE-trending (~030°), E/W-dipping, Late Triassic to Early Cretaceous syn-rift faults with strike lengths of 5–48 km; we model 5 listric faults with surface angles 45–70°, detachment depths 0.5–0.8 normalized, and displacements 0.4–0.7 × rupture_shift_scale. **(2) Subsidiary faults**—synthetic and antithetic structures subsidiary to the primary faults (Alrefaee et al., 2018); we model 500 (only 488 retained in final model) with displacements 16.7–30% of fault length and log-uniform length distribution (10–100% of primary average). This diversity provides an ideal template for training robust fault detection models.

*Table 5.* Fault population statistics.

| Property | Primary (5) | Subsidiary (500) |
|---|---|---|
| Azimuth | $\sim 30°/210°$ (NNE) | $\pm 10°$ from primary |
| Surface Angle | 45–70° | Similar to parent |
| Length | Fixed per template | Log-uniform (10–100% avg) |
| Displacement | 0.4–0.7 $\times$ shift_scale | 16.7–30% of length |
| Depth Extent | Through pre-fault section | Upper pre-fault section |

## B. Model Builder Implementation

### B.1. Architecture Overview

The Model Builder employs an event-based modular architecture. The core container (`GeoModel`) maintains three-dimensional arrays: P-wave velocity (float32), layer identifiers (int32), and fault labels (int32). Modules operate on normalized coordinates (0–1), enabling application to any sedimentary basin through parameter adjustment. Mathematical operations mirror real geological mechanisms: layer stacking follows stratigraphic superposition, fault displacement uses coordinate transformations matching extensional tectonics, and deformation modules apply rotatable strain fields consistent with compaction physics.

### B.2. Module Descriptions

*Table 6.* Model builder modules and their capabilities.

| Module | Key Features | Output |
|---|---|---|
| **Deposit** | Sublayer generation (variable-thickness beds); depth-dependent velocity gradients; sinc interpolation for profile morphing; 3D Perlin noise texture | Parallel, continuous reflectors with internal heterogeneity |
| **Fault** | Listric geometry (45–70° surface angle, detachment depth); rotated coordinate systems; displacement curves with lateral die-off; map-view curvature | Offset reflectors, fault plane reflections |
| **Squish** | Rotatable deformation fields; fractal amplitude modulation (4 octaves); configurable max shift (600–1600 m) | Antiforms/synforms, thickness variations |
| **Stratigraphic** | Pattern cycles (aggradation, progradation, retrogradation); sigmoid/tangential clinoform geometry; fractal roughness | Oblique reflectors (sequence stratigraphy) |
| **Erosion** | Sigmoid base surface; canyon perturbations; fractal bathymetric noise | Truncation surfaces, unconformities |

Table 6 shows some of the key procedural model building modules that were used to build the 3-D synthetic model that is a close representation to the Gorgon model.

### B.3. Construction Sequence

Table 7 details the 15-step construction sequence used to build the final velocity model. Each step applies a specific module with the listed parameters, following geological superposition from deep basement (step 1) through shallow water column (step 15).

### B.4. Test Cube Output Specifications

As detailed in Table 8, these properties define the generated velocity model, which was transformed into a pseudo-seismic cube for the purpose of fault detection.

*Table 7.* Model construction sequence (chronological order).

| Step | Event | Lithology | Velocity | Samples/Notes |
|------|-------|-----------|----------|---------------|
| 1 | Basement | — | — | Fractal continental shelf (not seen in published model) |
| 2 | Initial Deposit | Sandstone | 4200 m/s | 200 samples |
| 3 | Squish 1 | — | — | Max shift 1200 m |
| 4 | Layer 1 Deposit | Sandstone | 4075 m/s | 80 samples |
| 5 | Squish 2 | — | — | Max shift 800 m |
| 6 | Layer 2 Deposit | Shale | 3925 m/s | 200 samples |
| 7 | Squish 3 | — | — | Max shift 800 m |
| 8 | Layer 3 Deposit | Sandstone | 3775 m/s | 80 samples |
| 9 | Pre-Fault Deposit | Siltstone | 2600 m/s | 240 samples |
| 10 | Primary Faults | — | — | 5 listric normal |
| 11 | Secondary Faults | — | — | 488 support faults |
| 12 | Inclined Post-Fault | Siltstone | 2400 m/s | 40–80 samples |
| 13 | Stratigraphic Seq. | Carbonate | 2000–2200 m/s | 6 cycles |
| 14 | Erosion | — | — | 5% depth removal |
| 15 | Water Fill | Water | 1500 m/s | — |

*Table 8.* Output Specifications of 3-D Gorgon Synthetic Velocity Model and Pseudo-seismic cube.

| Property | Value |
|----------|-------|
| Dimensions | $1500 \times 1500 \times 1200$ voxels |
| Physical Size | 18.75 km $\times$ 18.75 km $\times$ 6.0 km |
| Voxel Spacing | 12.5 m $\times$ 12.5 m $\times$ 5.0 m |
| Total Voxels | 2.70 billion |
| Unique Layer Labels | 14 |
| Fault Labels | 493 (5 primary + 488 secondary) |
| Velocity Range | 1500–4912 m/s |

