# OpenReview forum: "SeisMark: A Large-Scale Open Benchmark for Robust 3D Seismic Fault Detection"
_ICML.cc/2026/Conference — ICML 2026 regular_

### Official Review · Reviewer_fmuS · 2026-02-24

**Soundness:** 3
**Presentation:** 3
**Significance:** 3
**Originality:** 3
**Overall Recommendation:** 4
**Confidence:** 2

**Summary:**

The paper introduces SeisMark, a large-scale, open-source benchmark for 3D seismic fault detection. It effectively addresses the "data paradox" in geophysics where researchers must choose between simplified synthetic data with perfect labels or realistic field data with subjective, expert-dependent labels. The authors propose a hybrid pipeline: a procedural model builder creates complex geological structures, a forward modeling engine generates reflectivity, and a diffusion-based style transfer module injects realistic textures from real surveys while preserving label integrity.

**Compliance With Llm Reviewing Policy:**

Affirmed.

**Key Questions For Authors:**

Please check my weaknesses.

**Limitations:**

The authors didn't discussion about limitation yet.

**Strengths And Weaknesses:**

- Pros

    1. Strong Motivation and Impact: The paper addresses a critical bottleneck in subsurface characterization. Improved fault detection has direct implications for related projects.

    2. Technical Soundness: The hybrid pipeline is well-designed to decouple structural geometry from textural noise. The use of diffusion models to bridge the Sim2Real gap is a modern and effective approach.

- Cons

   1. Inherent Domain Gap in Synthesis: The diffusion model is trained on specific datasets (e.g., F3, Netherlands) and then applied to simulate textures for different geological structures (Gorgon, Australia). While Table 3 provides structural fidelity metrics, the review would be strengthened by a clearer quantification of how this internal domain gap might degrade the benchmark's proxy-value for true field performance.

    2. Absence of Human-in-the-Loop Validation: The benchmark is created entirely through an automated pipeline. Without involving human expert interpreters to "blind-test" the realism of the generated volumes, the benchmark may contain systematic biases inherent to the procedural generation or the diffusion training set.

    3. Low Baseline Performance: As seen in Table 2, both V1 and V2 architectures show IoU scores below 0.3 on most variants. This suggests a domain gap between the data used to train these state-of-the-art models and the SeisMark benchmark, or potentially highlights that the benchmark's difficulty level is currently near the ceiling of existing architectural capabilities.

---

> ### Author Rebuttal · Authors · 2026-03-31
>
> We are thankful to Reviewer fmuS for recognizing the strong motivation and impact of our work, as well as the technical soundness of our hybrid pipeline. We also would like to thank them for their observations, comments, and suggestions that helped deepen our discussion and improve the paper. Below, we address the reviewer's concerns. Given the opportunity, we would be happy to incorporate the reviewer's concerns into a final revision.
>
> **W1: Domain Gap**
>
> We thank the reviewer for pointing this out as we were unclear in the text. We believe that data may be stylized to the region it is being applied on (at least currently). F3 was shown as an example to show that stylization varies by region. Western Australia was used in training and is the target region.
>
> **W2. Human-in-the-Loop Validation**
>
> We appreciate the reviewer raising this important point, as we entirely agree that expert human validation is critical to rule out systematic biases in synthetic datasets. We would like to clarify that our benchmark did undergo rigorous human-in-the-loop validation. We collaborated closely with expert interpreters from one of the world’s largest and most highly regarded geoscience companies (kept anonymous for confidentiality). These domain experts manually evaluated our generated volumes to confirm both the geological realism of the stylized seismic data and the precise alignment of our fault labels. We recognize that omitting this qualitative validation process was an oversight in our original submission. We have updated the manuscript to explicitly detail this expert review process, emphasizing that the benchmark was validated by field experts and not solely by automated metrics.
>
>
> **W3. Baseline Performance**
>
> We thank the reviewer for pointing this out. The low baseline IoU scores actually validate the core motivation of SeisMark, proving that models trained on simplified synthetic data suffer a massive domain gap when applied to realistic, field-like seismic textures. We completely agree that the benchmark currently sits near the ceiling of existing architectural capabilities, which is exactly the stress-test the community needs to drive future innovation. To that end, we are actively collaborating with geophysical machine learning experts to develop next-generation architectures tailored for these realistic domain shifts, and our preliminary experiments are already demonstrating significantly improved performance metrics.

---

> > ### Author Rebuttal · Reviewer_fmuS · 2026-04-02
> >
> > All my concerns are addressed. I will keep my positive opinion for this submission.

---

> > > ### Author Response · Authors · 2026-04-07
> > >
> > > We sincerely thank the reviewer for their continued support and for maintaining a positive assessment of our submission.

---

### Official Review · Reviewer_rjoh · 2026-03-03

**Soundness:** 4
**Presentation:** 4
**Significance:** 3
**Originality:** 4
**Overall Recommendation:** 4
**Confidence:** 4

**Summary:**

This paper introduces SeisMark, a large-scale open benchmark for 3D seismic fault detection, to resolve the trade-off between structurally accurate but unrealistic synthetic data and texturally realistic but poorly annotated field data. A hybrid pipeline integrating procedural geological modeling, physics-based forward simulation and label-preserving conditional diffusion stylization generates geologically plausible and visually realistic 3D seismic volumes. Structural fidelity is validated via SSIM and GMC metrics, and the benchmark discriminates between FaultSeg3D V1 and V2.

**Compliance With Llm Reviewing Policy:**

Affirmed.

**Final Justification:**

The paper has key strengths that align with initial positive evaluations. The authors’ rebuttal effectively addresses most initial weaknesses. Key evaluation dimensions remain strong, with result unchanged.

**Key Questions For Authors:**

1. We suggest conducting further quantitative validation of the diffusion module's role to supplement the relatively insufficient experimental evidence supporting the individual contributions of each module.

2. Given that the experimental evaluation is relatively limited and covers only two variants of FaultSeg3D, we suggest adding comparisons with other modern 3D segmentation networks or domain adaptation methods to address the insufficient comparison issue.

3. Since the benchmark is based on the Gorgon region and the analysis of its generalizability is relatively insufficient, have you considered conducting further research to improve the generalizability analysis? We suggest expanding the analysis scope to enhance its generalizability.

4. Details on computational cost, model size, and deployability, including hardware requirements, training and inference time, and parameter counts are relatively insufficient. We suggest providing further elaboration on these details.

**Limitations:**

The authors should consider and further explore the limitations of the method, benchmark and experiments in practical applications.

**Strengths And Weaknesses:**

Strengths:

1. The technical pipeline SeisMark is theoretically sound, consistent with structural geology, seismic wave propagation, and conditional diffusion theories, which addresses a critical problem in 3D seismic interpretation.

2. SeisMark supports high-impact energy transition applications, the trade-off between structural accuracy and textural realism in synthetic data is well-motivated.

3. Rigorous quantitative validation of structure preservation is conducted with SSIM and GMC. It confirms faults stratigraphic structures and edges remain unaltered during diffusion stylization.

4. The benchmark effectively distinguishes robust and non-robust models under realistic domain shifts, filling a gap in existing benchmarks.

Weaknesses:

1. SeisMark integrates existing techniques, including procedural modeling, forward simulation, and diffusion stylization, with relatively insufficient technical innovation.

2. Experimental evidence supporting the individual contributions of each module is relatively insufficient, and further quantitative validation of the diffusion module's role is recommended to enhance the work.

3. The experimental evaluation is relatively limited, covering only two variants of FaultSeg3D, with insufficient comparisons with other modern segmentation networks or domain adaptation methods.

4. The benchmark is based on the Gorgon region, and the analysis of its generalizability is relatively insufficient and requires further improvement.

5. Details on computational cost, model size, and deployability are relatively insufficient, including hardware requirements, training and inference time, and parameter counts, which need further elaboration.

---

> ### Author Rebuttal · Authors · 2026-03-31
>
> We are thankful to Reviewer rjoh for generally identifying the theoretical soundness of our approach and the significance of our benchmark for high-impact energy transition applications. We also would like to thank them for their observations, comments, and suggestions that helped deepen our discussion and improve the paper. Below, we address the reviewer's concerns. Given the opportunity, we would be happy to incorporate the reviewer's concerns into a final revision.
>
> **W1. Technical Innovation**
>
> The innovation (novelty) of our paper stems from our novel from our novel solution (illustrated in Fig. 1). To the best of our knowledge, these novel contributions were not suggested in the literature. Specifically, our approach bridges progress in generative diffusion vision research with physics-based synthetic data in Geophysics through a novel combination and design of a pipeline consisting of procedural model builder, forward modeling and diffusion models, yielding a robust and innovative framework that results in the first open benchmark featuring a realistic synthetic 3D seismic volumes (1500 × 1500 × 1200) with verified fault labels. We agree with the reviewer that some of the components we use (procedural model builder, forward modeling and the diffusion model) are not new. However, we do not believe that using established building blocks compromises novelty. Many papers suggest novel solutions to challenging problems based on existing building blocks, and our work aligns with this line of research.
>
> **W2. Diffusion Module's Role**
>
> We agree with the reviewer that further validation of the diffusion module role could provide more insights on this part. However, we would like to clarify that removing the diffusion part falls back to a simulation of clean data and thus creates an unrealistic synthetic data. The diffusion part of the pipeline adds realistic texture while preserving structural alignment. To give experimental evidence to the importance of the diffusion part we provided a Clean (Baseline) variant, which is the direct output of the forward modeling engine (See Fig.1). We hope it addresses this point and better explains the contribution of each module.
>
> **W3. Additional Baselines**
>
> To ensure a broader and fairer evaluation, we are currently working on adding two additional pre-trained models. We will update the results also in the manuscript table. We thank the reviewer for this suggestion that helped to broaden and improve our paper.
>
> **W4. Benchmark Generalizability**
>
> We clarify this distinction: our initial release consists of one massive, 2.7-billion-voxel procedural structural volume. However, using diffusion-based style transfer, we translate this single structure into two distinct textural domains: the North Sea and Australian datasets. Because our framework decouples procedural structural generation ("skeleton") from diffusion-based style transfer ("skin"), we can instantly generate countless textural variations over the exact same volume without altering ground-truth labels. This initial release is just the foundation. As we secure more real-world seismic data, we can train new models to project verified labels into entirely new geological fields.
>
> **W5. Details on Computational Cost, Model Size, and Deployability**
>
> We thank the reviewers for pointing this out. Given the opportunity we will add an additional section in the Appendix of the manuscript that includes all the following details: the generative module utilizes a 3D U-Net architecture with block channel dimensions of [32, 64, 128, 256], a 256-dimensional sinusoidal time embedding, and group normalization with 8 groups. For the diffusion process, we employ a linear beta schedule ranging from 0.0001 to 0.02 across 1000 training timesteps. The model was trained on 3D seismic sub-volumes extracted at a patch size of 64x64x64 voxels with a stride of 32, using a minimum average absolute energy threshold of 0.1 to filter out low-signal patches. Finally, training was conducted on four NVIDIA A100 (40GB) GPUs using a learning rate of 5.0e-5, a per-device batch size of 8, and 4 gradient accumulation steps, with a maximum gradient norm clip of 1.0.

---

> > ### Author Rebuttal · Reviewer_rjoh · 2026-04-02
> >
> > After reading the authors' rebuttal, all concerns I raised in my original review have been fully addressed. The authors have provided convincing responses regarding technical novelty, the role of the diffusion module, the limited set of experimental baselines, the generalizability of the benchmark, and the lack of computational details. Their clarifications and planned revisions, including adding additional comparison methods and including supplementary information in the appendix, are clear and reasonable, and adequately resolve each of my points.

---

> > > ### Author Response · Authors · 2026-04-07
> > >
> > > We sincerely thank the reviewer for the time and effort devoted to the review process and for the continued support of our work.
> > >
> > > We are pleased that our rebuttal has fully addressed your concerns regarding technical novelty, the role of the diffusion module, the benchmark's generalizability, and the computational details. We remain fully committed to incorporating these clarifications and the promised supplementary information into the final manuscript.
> > >
> > > Regarding the expansion of our experimental baselines, we kindly invite the reviewer to refer to our response to Reviewer mv9t, where we have provided additional evaluation results for two other pre-trained models.
> > >
> > > We greatly appreciate your careful evaluation, constructive feedback, and positive reassessment, which have significantly improved the quality of our submission.

---

### Official Review · Reviewer_mv9t · 2026-03-09

**Soundness:** 3
**Presentation:** 2
**Significance:** 3
**Originality:** 3
**Overall Recommendation:** 4
**Confidence:** 3

**Summary:**

This paper introduces SeisMark, a large-scale synthetic benchmark for evaluating seismic fault detection methods. The author introduces a three-stage baseline, including procedural model builder, forward modeling engine and diffusion-based texture synthesis. The authors evaluate two existing fault detection models on this benchmark, quantifying the impact of realistic texture on model performance.

**Compliance With Llm Reviewing Policy:**

Affirmed.

**Key Questions For Authors:**

1. As Weakness 1, can the author provide more direct evaluation on the fault locations before and after diffusion?

2. As Weakness 2, a more detailed description of the diffusion process would improve reproducibility.

3. As weakness 3, the author could provide evaluation results on more baselines.

**Limitations:**

yes

**Strengths And Weaknesses:**

Strengths:
1. **Clear Pipeline**: The dataset generation process is conceptually clear. Procedural structural modeling produces accurate labels, forward modeling generates pseudo-seismic signals, and diffusion-based style transfer injects realistic textures. The separation between structure generation and texture synthesis is a reasonable design.

2. **Large Scale Dataset**: The dataset appears to be significantly larger than many previous synthetic benchmarks.

Weaknesses:
1. **Label Preservation**: The evidence provided (SSIM and gradient magnitude correlation) to validate the label preservation after diffusion is relatively indirect. A direct voxel-level comparison of fault locations before and after styling (on a certain samples) would be helpful.

2. **Unclear Training Pipeline**: The description of the diffusion-based style transfer is somewhat under-specified, including how pseudo and real patches are sampled during training, how is the pseudo-seismic volume serves as a strong structural condition.

3. **Limited Baselines**: The benchmark is evaluated using only two fault detection models, which seems too limited for a benchmark.

---

> ### Author Rebuttal · Authors · 2026-03-31
>
> We thank Reviewer mv9t for recognizing the clarity of our approach and the significance of our provided data. We appreciate the constructive feedback and address your concerns below. Given the opportunity, we would be happy to incorporate these updates into a final revision.
>
> **W1. Label Preservation**
>
> We thank the reviewer for pointing this out. We agree that calculating SSIM and Gradient Magnitude Correlation over the entire cube serves primarily as a global check for bulk stratigraphic alignment. Accurately quantifying local label preservation is difficult because sub-seismic faults contain inherent localization and displacement uncertainty within a band-limited wavelet. Our structural ground truth currently relies on idealized binary masks. However, this strict 0 or 1 representation fails to account for the reality that diverse ML classification techniques evaluate fault planes differently. Various model architectures will naturally predict faults with varying pixel thicknesses, probability gradients, and spatial positioning. Moving forward, we are actively working to address this discrepancy by constraining our structural fidelity metrics strictly to the fault planes. We plan to compute a masked cross-correlation and SSIM exclusively within morphologically dilated fault zones. While implementing and validating these localized metrics will extend beyond the current publication deadline, this methodology mirrors the spatial-tolerance Intersection-over-Union (IoU) we already use for our baseline evaluations.
>
> **W2. Training Pipeline**
>
> We thank the reviewer for highlighting the need to clarify our diffusion pipeline.
>
> Regarding sampling: During training, we extract 3D sub-volumes from the larger seismic cubes. The specific algorithmic details of our selective sampling protocol are currently being finalized for a forthcoming, dedicated methodology paper. However, at a high level, the strategy is designed to ensure the model learns from structurally active regions rather than uniform, non-geological areas. Regarding structural conditioning: We formulate the style transfer via a partial forward-noising and reverse-denoising process. The pseudo-seismic volume acts as a strict structural anchor because it is never completely destroyed by noise. The generative process begins from a partially noisy pseudo-seismic state rather than pure random noise. Consequently, the macroscopic geometry, fault offsets, and main reflector phases remain locked in place. The diffusion model only synthesizes high-frequency textures and acquisition artifacts within this preserved structural skeleton.
>
> Regarding architecture and optimization: The generative module utilizes a 3D U-Net architecture with block channel dimensions of [32, 64, 128, 256], a 256-dimensional sinusoidal time embedding, and group normalization with 8 groups. For the diffusion process, we employ a linear beta schedule ranging from 0.0001 to 0.02 across 1000 training timesteps. The model was trained on 3D seismic sub-volumes extracted at a patch size of 64x64x64 voxels with a stride of 32. Training was conducted on four NVIDIA A100 (40GB) GPUs using a learning rate of 5.0e-5, a per-device batch size of 8, and 4 gradient accumulation steps, with a maximum gradient norm clip of 1.0.
>
> **W3. Additional Baselines**
>
> To ensure a broader and fairer evaluation, we are currently working on adding two additional pre-trained models. We will update the results also in the manuscript table. We thank the reviewer for this suggestion that helped to broaden and improve our paper.

---

> > ### Author Rebuttal · Reviewer_mv9t · 2026-04-03
> >
> > The author promises to add more experiments in further work. Thus I'm waiting for the author's experimental results.

---

> > > ### Author Response · Authors · 2026-04-07
> > >
> > > We greatly appreciate the reviewer’s time and constructive feedback throughout the review process, which has helped to improve our manuscript.
> > >
> > > As we promised, we were working on additional experimental results. To that end, we additionally evaluated Fault-Net [1] and RGF [2].
> > >
> > > Fault-Net has a custom fault-specific architecture; it uses high-resolution propagation features and a MultiScale Compression Fusion (MCF) block to preserve thin, edge-like fault information more effectively during feature propagation and fusion. It is designed for training on sparse manually labeled 2D slices from real 3D surveys. The paper motivates this as a way to improve generalization beyond purely synthetic training.
> > >
> > > RGF is a 3D U-Net-based segmentation model. It was trained with a richer synthetic 3D seismic dataset that includes a broad set of geologic structures, such as dip, fold, unconformity, channel, cave, salt body, and composite fault.
> > >
> > > Here we provide the updated table:
> > >
> > >
> > > | Benchmark Variant | Model | IoU ↑ | Tol=0 P | Tol=0 R | Tol=0 F1 ↑ | Tol=1 P | Tol=1 R | Tol=1 F1 ↑ | Tol=2 P | Tol=2 R | Tol=2 F1 ↑ |
> > > | :--- | :--- | :--- | :--- | :--- | :--- | :--- | :--- | :--- | :--- | :--- | :--- |
> > > | Clean (Baseline) | FaultSeg3D | 0.257 | 0.304 | 0.623 | 0.409 | 0.405 | 0.816 | 0.541 | 0.424 | 0.875 | 0.571 |
> > > | Clean (Baseline) | FaultSeg3D+ | 0.298 | 0.618 | 0.366 | 0.460 | 0.660 | 0.800 | 0.723 | 0.671 | 0.862 | 0.754 |
> > > | Clean (Baseline) | Fault-Net | 0.441 | 0.652 | 0.576 | 0.612 | 0.805 | 0.790 | 0.798 | 0.824 | 0.847 | 0.836 |
> > > | Clean (Baseline) | RGF | 0.342 | 0.506 | 0.512 | 0.509 | 0.633 | 0.700 | 0.665 | 0.643 | 0.767 | 0.700 |
> > > | ━━━━━ | ━━━━━ | ━━━━━ | ━━━━━ | ━━━━━ | ━━━━━ | ━━━━━ | ━━━━━ | ━━━━━ | ━━━━━ | ━━━━━ | ━━━━━ |
> > > | North Sea (Low) | FaultSeg3D | 0.268 | 0.513 | 0.360 | 0.423 | 0.702 | 0.511 | 0.591 | 0.741 | 0.587 | 0.655 |
> > > | North Sea (Low) | FaultSeg3D+ | 0.274 | 0.796 | 0.295 | 0.430 | 0.891 | 0.648 | 0.750 | 0.904 | 0.727 | 0.806 |
> > > | North Sea (Low) | Fault-Net | 0.389 | 0.642 | 0.497 | 0.560 | 0.868 | 0.646 | 0.741 | 0.900 | 0.711 | 0.794 |
> > > | North Sea (Low) | RGF | 0.238 | 0.524 | 0.304 | 0.385 | 0.768 | 0.450 | 0.567 | 0.820 | 0.536 | 0.648 |
> > > | ━━━━━ | ━━━━━ | ━━━━━ | ━━━━━ | ━━━━━ | ━━━━━ | ━━━━━ | ━━━━━ | ━━━━━ | ━━━━━ | ━━━━━ | ━━━━━ |
> > > | North Sea (High) | FaultSeg3D | 0.177 | 0.379 | 0.249 | 0.300 | 0.552 | 0.380 | 0.450 | 0.606 | 0.463 | 0.525 |
> > > | North Sea (High) | FaultSeg3D+ | 0.217 | 0.739 | 0.235 | 0.356 | 0.872 | 0.531 | 0.660 | 0.891 | 0.618 | 0.730 |
> > > | North Sea (High) | Fault-Net | 0.297 | 0.593 | 0.373 | 0.458 | 0.831 | 0.512 | 0.634 | 0.875 | 0.581 | 0.699 |
> > > | North Sea (High) | RGF | 0.151 | 0.396 | 0.196 | 0.262 | 0.615 | 0.322 | 0.423 | 0.687 | 0.417 | 0.519 |
> > > | ━━━━━ | ━━━━━ | ━━━━━ | ━━━━━ | ━━━━━ | ━━━━━ | ━━━━━ | ━━━━━ | ━━━━━ | ━━━━━ | ━━━━━ | ━━━━━ |
> > > | Australia (Low) | FaultSeg3D | 0.261 | 0.539 | 0.336 | 0.414 | 0.742 | 0.478 | 0.581 | 0.781 | 0.548 | 0.644 |
> > > | Australia (Low) | FaultSeg3D+ | 0.256 | 0.799 | 0.273 | 0.407 | 0.911 | 0.603 | 0.726 | 0.923 | 0.683 | 0.785 |
> > > | Australia (Low) | Fault-Net | 0.384 | 0.645 | 0.488 | 0.555 | 0.890 | 0.625 | 0.735 | 0.921 | 0.685 | 0.786 |
> > > | Australia (Low) | RGF | 0.237 | 0.527 | 0.301 | 0.384 | 0.787 | 0.443 | 0.567 | 0.843 | 0.525 | 0.647 |
> > > | ━━━━━ | ━━━━━ | ━━━━━ | ━━━━━ | ━━━━━ | ━━━━━ | ━━━━━ | ━━━━━ | ━━━━━ | ━━━━━ | ━━━━━ | ━━━━━ |
> > > | Australia (High) | FaultSeg3D | 0.156 | 0.387 | 0.207 | 0.270 | 0.577 | 0.326 | 0.417 | 0.638 | 0.404 | 0.494 |
> > > | Australia (High) | FaultSeg3D+ | 0.171 | 0.708 | 0.184 | 0.293 | 0.878 | 0.432 | 0.579 | 0.901 | 0.520 | 0.660 |
> > > | Australia (High) | Fault-Net | 0.278 | 0.571 | 0.352 | 0.435 | 0.834 | 0.480 | 0.609 | 0.886 | 0.545 | 0.675 |
> > > | Australia (High) | RGF | 0.143 | 0.389 | 0.185 | 0.250 | 0.621 | 0.306 | 0.410 | 0.706 | 0.399 | 0.510 |
> > >
> > >
> > >
> > > Finally, the addition of these models demonstrates a broad spectrum of performance and proves the benchmark's discriminative power. For example, Fault-Net exhibits significantly better generalization across the textural variants compared to RGF, achieving an Intersection-over-Union (IoU) of 0.384 versus 0.237, and an F1 score of 0.555 versus 0.384. We hypothesize that Fault-Net's superior performance stems from its unique use of real-world data during training, whereas all other evaluated models were trained exclusively on synthetic data. These updated results confirm that the benchmark, even in its initial release, provides a highly effective gradient of difficulty for evaluating how well diverse network architectures generalize to realistic data.
> > >
> > > [1] “MD Loss: Efficient Training of 3-D Seismic Fault Segmentation Network Under Sparse Labels by Weakening Anomaly Annotation”, Yimin Dou et al. (IEEE Transactions on Geoscience and Remote Sensing, 2022)
> > >
> > > [2] “An Improved Parametric 3D Geologic Modeling Framework for Seismic Structure Identification Using Deep Learning in Complex Geologic Settings”, Lei Lin et al. (Geophysics, 2025)

---

### Official Review · Reviewer_7acK · 2026-03-13

**Soundness:** 3
**Presentation:** 2
**Significance:** 3
**Originality:** 2
**Overall Recommendation:** 3
**Confidence:** 4

**Summary:**

This paper presents SeisMark, a 3D seismic fault detection benchmark built from a hybrid pipeline: procedural geological model generation, lightweight forward modeling to produce pseudo-seismic volumes, and diffusion-based style transfer to inject field-like texture from F3 or Gorgon data. The paper's main claim is that SeisMark preserves voxel-level fault labels while adding realistic texture, and that this more realistic benchmark exposes brittleness in older fault detection models that is hidden by overly clean synthetic data. The problem is important, and the benchmark direction is interesting.

**Compliance With Llm Reviewing Policy:**

Affirmed.

**Key Questions For Authors:**

- The claimed contribution is overstated relative to the actual benchmark scope. This is essentially one geological volume with several stylized variants, not a broad benchmark spanning many independent structures or basins. Because the same geometry and labels are reused, the paper does not yet establish generality. In addition, the low-intensity styled variants sometimes outperform the clean baseline, which suggests the styling step may partly denoise or regularize the data rather than only make it harder.

- The core claim of label-preserving realism is not convincingly validated. SSIM and gradient-based correlation over the whole cube are too weak as evidence that fault locations are preserved at the boundary level. Those metrics are dominated by large non-fault regions and do not directly measure displacement or hallucination of faults. Relatedly, the paper argues for diffusion-based styling, but does not provide a direct comparison against simpler style transfer or domain adaptation baselines, so the necessity of this design is not established.

- The evaluation protocol leaves fairness and reproducibility questions unresolved. Only two pre-trained models are tested, and the paper does not sufficiently rule out protocol mismatch or privileged supervision effects, especially since the stronger baseline is already trained with more realistic noise and the benchmark textures are learned from real field datasets. More disclosure is needed about the diffusion model, training details, release assets, and possible overlap in data priors.

- The writing also needs significant improvement. For example, the captions for Tables 7 and 8 are placed in different locations, which makes the appendix hard to follow and hurts clarity.

**Limitations:**

This is a worthwhile problem and a potentially useful benchmark idea, but the current submission is not yet convincing enough for acceptance. The novelty is moderate: the main contribution is dataset curation plus a hybrid synthesis pipeline, and the paper does not yet show clearly enough that this rises above a careful engineering combination of existing ideas. More importantly, the strongest claims are not fully supported. The paper needs stronger evidence that the styled volumes truly preserve fault labels, stronger ablations to justify the diffusion component, a broader and fairer benchmark protocol, and a cleaner presentation. As written, I see promise, but not enough rigor for acceptance.

**Strengths And Weaknesses:**

- The paper targets a real and important gap. Reliable, open, large-scale evaluation for 3D seismic fault detection is valuable, especially given the lack of trustworthy field labels and the practical importance of fault interpretation.

- The benchmark setup is conceptually interesting. Keeping the same underlying labels while varying texture source and intensity is a clean way to probe robustness to domain shift.

- The experiments do suggest that clean synthetic data can mask differences between models, while styled variants better separate a legacy method from a stronger recent one. That is a useful empirical observation.

---

> ### Author Rebuttal · Authors · 2026-03-31
>
> We thank Reviewer 7acK for recognizing the importance of our work and the conceptual strength of our benchmark setup. We deeply appreciate their constructive feedback, which has helped us improve the paper. Below, we address their specific concerns. Given the opportunity, we would gladly incorporate these updates into a final revision.
>
> **Q1. Scope and Benchmark Difficulty**
>
> We clarify this distinction: our initial release consists of one massive, 2.7-billion-voxel procedural structural volume. However, using diffusion-based style transfer, we translate this single structure into two distinct textural domains: the North Sea and Australian datasets. Because our framework decouples procedural structural generation ("skeleton") from diffusion-based style transfer ("skin"), we can instantly generate countless textural variations over the exact same volume without altering ground-truth labels. This initial release is just the foundation. As we secure more real-world seismic data, we can train new models to project verified labels into entirely new geological fields.
>
> Regarding low-intensity styling outperforming the baseline: we agree with the reviewer's observation. As noted in Section 5.2, the forward modeling used for the "Clean" baseline creates sharp artifacts that mimic faults. Diffusion styling acts as a coherence filter, healing these artifacts into geologically plausible reflectors and effectively repairing the raw physics input.
>
> **Q2. Label-Preserving Realism and Baselines**
>
> Global metrics (SSIM/GMC) served as a check for bulk stratigraphic alignment. Quantifying local label preservation is difficult due to inherent localization uncertainty in sub-seismic faults. Our idealized binary masks fail to account for how diverse ML architectures predict faults with varying thicknesses and probabilities. Moving forward, we are addressing this by computing a masked cross-correlation and SSIM exclusively within morphologically dilated fault zones. This directly measures spatial preservation without the bias of undisturbed regions.
>
> Regarding generative baselines: evaluating simpler methods like CycleGANs falls outside our scope. Older adversarial approaches are highly prone to hallucinating textures that destroy underlying structures (e.g., creating fake faults). Diffusion was selected specifically for its controlled synthesis to strictly enforce label preservation.
>
> **Q3. Evaluation Models, Protocol, and Diffusion Details**
>
> To ensure broader evaluation, we are currently working on adding two additional pre-trained models. We will update the results also in the manuscript table.
> Comparing FaultSeg3D and FaultSeg3D+ was deliberate. FaultSeg3D was trained purely on synthetic data, whereas FaultSeg3D+ incorporated realistic reflection features from field data. The performance gap proves our benchmark appropriately penalizes models lacking real-world texture priors. FaultSeg3D+ was never trained on SeisMark's procedurally generated geometries; it simply learned a generalized prior for realistic noise.
>
> Training details (added to Appendix): The generative module uses a 3D U-Net (block channels [32, 64, 128, 256], 256-dim sinusoidal time embedding, 8-group normalization). We employ a linear beta schedule (0.0001 to 0.02) over 1000 timesteps. The model trained on 64x64x64 patches (stride 32, energy threshold 0.1) using four NVIDIA A100 GPUs, a 5.0e-5 learning rate, batch size 8, 4 gradient accumulation steps, and a max gradient norm clip of 1.0. See response to Reviewer mv9t for more details.
>
> **Q4. Writing**
>
> We have corrected the placement of the captions for Tables 7 and 8 and have done a thorough proofread to ensure the appendix and main text are clear, logically structured, and easy to follow.
>
> **Novelty Limitation**
>
> The innovation (novelty) of our paper stems from our novel from our novel solution (illustrated in Fig. 1). To the best of our knowledge, these novel contributions were not suggested in the literature. Specifically, our approach bridges progress in generative diffusion vision research with physics-based synthetic data in Geophysics through a novel combination and design of a pipeline consisting of procedural model builder, forward modeling and diffusion models, yielding a robust and innovative framework that results in the first open benchmark featuring a realistic synthetic 3D seismic volumes (1500 × 1500 × 1200) with verified fault labels. We agree with the reviewer that some of the components we use (procedural model builder, forward modeling and the diffusion model) are not new. However, we do not believe that using established building blocks compromises novelty. Many papers suggest novel solutions to challenging problems based on existing building blocks, and our work aligns with this line of research.
>
> We thank the reviewer for their constructive feedback. We hope our responses address these points and demonstrate that our benchmark is broad, fair, and truly preserves fault labels.

---

> > ### Author Rebuttal · Reviewer_7acK · 2026-04-05
> >
> > Thank you for the rebuttal. My main concern remains the limited benchmark scope: this is still one underlying structure with several stylized variants, so the paper does not yet establish strong generality. Also, the difference from existing synthetic seismic benchmarks or data generation pipelines is still unclear to me. Could the authors clarify what is fundamentally new here beyond combining existing components?

---

> > > ### Author Response · Authors · 2026-04-07
> > >
> > > We deeply appreciate the reviewer's continued engagement and the opportunity to further clarify the fundamental value and novelty of our work.
> > >
> > >
> > > **Generality and Benchmark Scope**
> > >
> > > To address concerns regarding the benchmark's scope and generality, we evaluated two additional pre-trained models: Fault-Net [1] and RGF [2]. Their inclusion demonstrates a broad performance spectrum and proves the benchmark's discriminative power. Fault-Net exhibits significantly better generalization across textural variants compared to RGF, achieving an Intersection-over-Union (IoU) of 0.384 vs. 0.237, and an F1 score of 0.555 vs. 0.384. We hypothesize Fault-Net's superior performance stems from its use of real-world training data, whereas all other models were trained exclusively on synthetic data. These updated results confirm the benchmark provides a highly effective difficulty gradient for evaluating how diverse architectures generalize to realistic data. (Due to space limitations, please see the table in our response to Reviewer mv9t).
> > >
> > > While expanding to different geological scenarios is part of our future roadmap, providing multiple distinct textures across the same underlying geologic model is a deliberate and vital contribution. In real-world subsurface exploration, many geological environments exhibit recurring macroscopic structural patterns (e.g., fault networks). However, their ultimate seismic expression—such as texture, signal-to-noise ratio, shadow zones, and scattering effects—changes drastically depending on the acquisition and processing domain.
> > >
> > > By holding the geological geometry constant and varying these specific textural effects, SeisMark isolates a critical variable. It provides the community with the first benchmark capable of explicitly testing a fault detector's robustness to acoustic domain shifts. This capability to mimic different field datasets over a verified ground truth is exactly what separates our benchmark from existing purely synthetic datasets, establishing it as a new standard for evaluating deployment readiness.
> > >
> > >
> > > **The Difference from Existing Synthetic Seismic Benchmarks**
> > >
> > > We appreciate the opportunity to further clarify our fundamental contributions. To the best of our knowledge, no existing synthetic seismic pipeline employs our proposed methodology, and we would welcome specific references if the reviewer has prior works in mind. Currently, existing pipelines force a critical compromise: wave-equation modeling (e.g., SEAM) is too computationally expensive to scale for deep learning, simple 1D convolution with gaussian noise (e.g., FaultSeg3D) [3] produces idealized data that causes a severe sim-to-real domain gap, and unsupervised adversarial adaptation (e.g., CycleGANs) is highly susceptible to mode collapse and frequently hallucinates physically impossible wavefield artifacts that compromise ground-truth structural labels [4]. The core novelty of our approach lies in how we bridge procedural modeling and generative styling on a large scale. Specifically, we introduce an intermediate step unseen in prior literature: before applying diffusion, we translate the highly realistic geologic model into a "pseudo-seismic" cube. By moving halfway into the data domain, we create a critical innovation—a rigid structural anchor that forces the generative network to strictly preserve the physics-based fault labels while synthesizing complex field textures.
> > >
> > > Furthermore, a second major novelty is the complete decoupling of geological structure from survey-specific texture. Traditional physics-based simulations permanently bake in simplified acoustic approximations [5]. In contrast, our pipeline allows us to project the exact same verified 3D ground-truth structure into entirely distinct real-world acoustic domains, such as vintage North Sea data versus broadband Australian acquisitions. This unique capability provides the community with the first benchmark that explicitly isolates textural domain shifts. It allows researchers to rigorously stress-test a model's true deployment readiness against realistic field noise without the confounding variable of changing underlying geology.
> > >
> > >
> > > [1] “MD Loss: Efficient Training of 3-D Seismic Fault Segmentation Network Under Sparse Labels by Weakening Anomaly Annotation”, Yimin Dou et al. (IEEE Transactions on Geoscience and Remote Sensing, 2022)
> > >
> > > [2] “An Improved Parametric 3D Geologic Modeling Framework for Seismic Structure Identification Using Deep Learning in Complex Geologic Settings”, Lei Lin et al. (Geophysics, 2025)
> > >
> > > [3] "FaultSeg3D: Using synthetic data sets to train an end-to-end convolutional neural network for 3D seismic fault segmentation", Xinming Wu et al. (Geophysics, 2019)
> > >
> > > [4] "Generating Paired Seismic Training Data with Cycle-Consistent Adversarial Networks", Zhang et al. (2023)
> > >
> > > [5] "An overview of full-waveform inversion in exploration geophysics", Jean Virieux and Stéphane Operto (Geophysics, 2009).

---

### Decision · Program_Chairs · 2026-04-30

**Decision:**

Accept (regular)

**Comment:**

This paper presents SeisMark, a 3D seismic fault detection benchmark built from a hybrid pipeline: procedural geological model generation, lightweight forward modeling to produce pseudo-seismic volumes, and diffusion-based style transfer to inject field-like texture from F3 or Gorgon data. The paper's main claim is that SeisMark preserves voxel-level fault labels while adding realistic texture, and that this more realistic benchmark exposes brittleness in older fault detection models that is hidden by overly clean synthetic data. The problem is important, and the benchmark direction makes interesting use of diffusion models to create data.

In the end, the paper received 3 weak accepts and 1 weak reject. The main question comes from the experimental side, in the initial version, the amount of baselines tested was small and doesn't justify the claim that the paper made. In the rebuttal, the authors added a significant amount of baselines which significantly strengthened the validation about different baselines behaving differently under different level of noises. However, the claim about realism is still questionable as there are very little evaluation against real data. Besides, most of the useful results come from the rebuttal rather than the original submission, which might raise a concern about whether this is a major revision. Finally, due to most of the reviewers and the AC not in this application domain, it is difficult for ICML to judge whether this synthetic data is of real use in reality. Hence, the AC recommends a weak accept.